# Volumetric extrusive rates of silicic supereruptions from the Afro-Arabian large igneous province

Jennifer E. Thines [1✉], Ingrid A. Ukstins [2], Corey Wall[3] & Mark Schmitz[3]

The main phase of silicic volcanism from the Afro-Arabian large igneous province preserves some of the largest volcanic eruptions on Earth, with six units totaling >8,600 km$^3$ dense rock equivalent (DRE). The large volumes of rapidly emplaced individual eruptions present a case study for examining the tempo of voluminous silicic magma generation and emplacement. Here were report high-precision $^{206}$Pb/$^{238}$U zircon ages and show that the largest sequentially dated eruptions occurred within 48 ± 34 kyr (29.755 ± 0.023 Ma to 29.707 ± 0.025 Ma), yielding the highest known long-term volumetric extrusive rate of silicic volcanism on Earth. While these are the largest known sequential silicic supereruptions, they did not cause major global environmental change. We also provide a robust tie-point for calibration of the geomagnetic polarity timescale by integrating $^{40}$Ar/$^{39}$Ar data with our $^{206}$Pb/$^{238}$U ages to yield new constraints on the duration of the C11n.1r Subchron.

[1] Department of Earth and Environmental Sciences, University of Iowa, 115 Trowbridge Hall, Iowa City, IA 52242, USA. [2] School of Environment, The University of Auckland, Private Bag 92 019, Auckland, New Zealand. [3] Department of Geosciences, Boise State University, 1295 University Drive, Boise, ID 83706, USA. ✉email: jennifer-thines@uiowa.edu

Many of the largest silicic eruptions on Earth occur in large igneous provinces (LIPs), with total eruptive volumes often exceeding 1000 km³ dense rock equivalent (DRE) for individual events (e.g., ~132 Ma Paraná-Etendeka, ~30 Ma Afro-Arabia, ~1.6 Ga Gawler Range), which are likely to be emplaced in rapid succession[1–3]. Although LIPs are generally considered to represent the most productive magmatic systems on Earth[4], uncertainty about volume estimates and imprecise or inaccurate age data for individual events preclude robust estimates of magma flux and volcanic output[1,5]. The silicic component of LIPs is largely understudied relative to their mafic counterpart and long-term rates of silicic magma generation have important implications for the energy budget and thermal evolution of the Earth. The Northern Yemen section of the Afro-Arabian LIP is an ideal testbed for using high-precision $^{206}Pb/^{238}U$ zircon dating to quantify the long-term volumetric extrusive rate of a series of flood volcanic eruptions, with three silicic supereruptions ($10^{15}$ kg or ~450 km³ of magma[6,7]) occurring within a 70–310 kyr timeframe at ca. 29.7 Ma[8–10].

Oligocene volcanism in Northern Yemen (Fig. 1) has been divided into three phases based on field observations, whole-rock geochemical correlations, and $^{40}Ar/^{39}Ar$ dating[8–10]: Main Basalts (31–29.7 Ma), Main Silicics (29.7–29.5 Ma), and Upper Bimodal (29.6–27.7 Ma). The Main Basalts phase is characterized by effusive basaltic volcanism and volumetrically represents 60–70% of the total erupted volume of Afro-Arabian lavas[9,11]. The Main Silicics phase saw the rapid emplacement of seven silicic pyroclastic units and the Upper Bimodal phase includes small-volume basaltic and rhyolitic eruptions[9]. The Northern Yemen section has excellent exposure and well-characterized stratigraphic relationships from field mapping, paleomagnetic studies, and correlations with distal deep sea co-ignimbrite ash deposits[8,9,12], while the Ethiopian section has been extensively faulted from active rifting with significant erosion around the volcanic plateau margin[13].

We focus on the Main Silicics phase, which contains some of the largest known silicic eruptions on Earth, with an estimated minimum total eruptive volume of ~8600 km³ DRE emplaced in present-day Yemen and Ethiopia over a period from 29.7 to 29.5 Ma[1,9]. Volcano–stratigraphic correlations in Yemen[9] suggest the emplacement of the Jabal Kura'a Ignimbrite (1600 km³ DRE; ~29.6 Ma) and Escarpment Ignimbrite (360 km³ DRE; ~29.6 Ma) was followed by a brief period of subsidence and erosion and then the rapid emplacement of the Green Tuff (60 km³ DRE; $^{40}Ar/^{39}Ar$ age = 29.59 ± 0.12 Ma[8]; Fig. 2), SAM Ignimbrite (2300 km³ DRE; $^{40}Ar/^{39}Ar$ age = 29.47 ± 0.14 Ma[10], Sana'a Ignimbrite (1600 km³ DRE; ~29.5 Ma; Fig. 2), and Iftar Alkalb caldera collapse mega-breccia (2700 km³ DRE; $^{40}Ar/^{39}Ar$ age = 29.48 ± 0.13 Ma[8]; Fig. 2). The Green Tuff has been interpreted as representing the initial airfall deposit preceding the emplacement of the SAM Ignimbrite based on the sharp upper contact between the units with no evidence of a time gap during emplacement[9]. These bracketed $^{40}Ar/^{39}Ar$ ages indicate that all four units, with a cumulative estimated minimum total eruptive volume of ~6700 km³ DRE, were emplaced in rapid succession within a timeframe of 70–310 kyr[8–10], but there are no robust estimates of magma generation rates or magma flux over this time interval.

Previous paleomagnetism and $^{40}Ar/^{39}Ar$ studies[8,9] indicate that the Main Silicics phase eruptions are a set of normal to reversed polarity units that encompass the duration of the C11n.1r Subchron, although overlapping ages for individual eruptions, due to analytical uncertainties, are currently unable to distinguish between the geomagnetic polarity time scale (GPTS) of Cande and Kent[14] and Huestis and Acton[15]. While there are several cooling events identified in the Oligocene $\delta^{18}O$ and $\delta^{13}C$ chemostratigraphy[16,17], the uncertainties of these ages also hinder the correlation of the Afro-Arabian silicic eruptions to any isotopic perturbations. In contrast to existing $^{40}Ar/^{39}Ar$ ages, the 0.1% precision of state-of-the-art chemical abrasion thermal ionization mass spectroscopy (CA-TIMS) U-Pb ages of zircons[18] can distinguish between the ages of these units outside analytical uncertainty. These new high-precision $^{206}Pb/^{238}U$ zircon ages are crucial to quantifying the rapid emplacement of voluminous Afro-Arabian silicic magmas in order to understand the transient nature of silicic supereruptions, demonstrating that these eruptions had little to no observed impact on long-term climate change, and constraining the duration of the C11n.1r Subchron.

## Results

**Zircon morphology**. Zircon crystals from the Escarpment, SAM and Sana'a Ignimbrites, and Iftar Alkalb were analyzed by cathodoluminescence (CL) imaging and laser ablation inductively coupled mass spectrometry (LA-ICP-MS) in order to distinguish petrochemical populations prior to CA-TIMS dating. The Escarpment Ignimbrite contains elongate prismatic crystals (typically 50–120 µm in length and, rarely, up to 150 µm) and smaller equant crystals (50–75 µm in length). Some prismatic crystals have oscillatory zoning with U-rich non-luminescent cores (CL dark). The SAM Ignimbrite contains elongate prismatic crystals that are both smaller (30–75 µm, rarely up to 125 µm) and less numerous than those found in the Escarpment Ignimbrite. Few crystals have subtle oscillatory zoning and one larger crystal ~120 µm in length has a non-luminescent, oscillatory zoned core with a lighter overgrowth rim. Crystals in the SAM Ignimbrite have a weakly paramagnetic behavior, likely due to abundant Fe-Ti oxide and apatite inclusions. The Sana'a Ignimbrite contains small elongate prismatic crystals (30–75 µm) with subtle to no oscillatory zoning. Zircon is abundant in Iftar Alkalb as anhedral to euhedral elongate prismatic and equant crystals that range in length from 30 to 120 µm. Internal morphologies are variable in Iftar Alkalb with populations of non-luminescent and luminescent zircon crystals with no oscillatory zoning, crystals with non-luminescent cores and lighter rims, and a few crystals with strong oscillatory zoning (see Supplementary Information for CL images).

In total, 273 laser ablation spot analyses were conducted on 79 crystals from the Escarpment Ignimbrite, 46 crystals from the SAM Ignimbrite, 31 crystals from the Sana'a Ignimbrite, and 95 crystals from Iftar Alkalb to identify xenocrysts (crystals that are several million years older than the relevant magma pulse and considered unrelated to the magma system[19]) and antecrysts (crystals that grew earlier and were incorporated in a later pulse[19,20]). The median uncertainty of a single LA-ICP-MS $^{206}Pb/^{238}U$ spot analysis is 3 Ma, too imprecise to distinguish antecryst populations for this magmatic system but adequate to determine older xenocrystic zircon crystals. Every unit except the Escarpment Ignimbrite contains >10% zircon crystals with LA-ICP-MS $^{206}Pb/^{238}U$ ages >33 Ma. The Sana'a Ignimbrite and Iftar Alkalb contain significant proportions of older zircons (30 and 29%, respectively), although in the Sana'a Ignimbrite this may be due to the low sample number ($n = 31$). There is no correlation between age and trace element (U, Th, Y, HREE) concentrations. CL dark zircon crystals in the Escarpment Ignimbrite and Iftar Alkalb have among the highest HREE concentrations and europium anomalies (Eu/Eu*) in each respective unit and the ages of the cores and rims of the few zircon crystals with clear zonation were indistinguishable outside uncertainty (Supplementary Data). The evolution of Eu/Eu* in zircons from the Escarpment, SAM and Sana'a Ignimbrites, and Iftar Alkalb requires 50–60% fractional crystallization of feldspar to produce

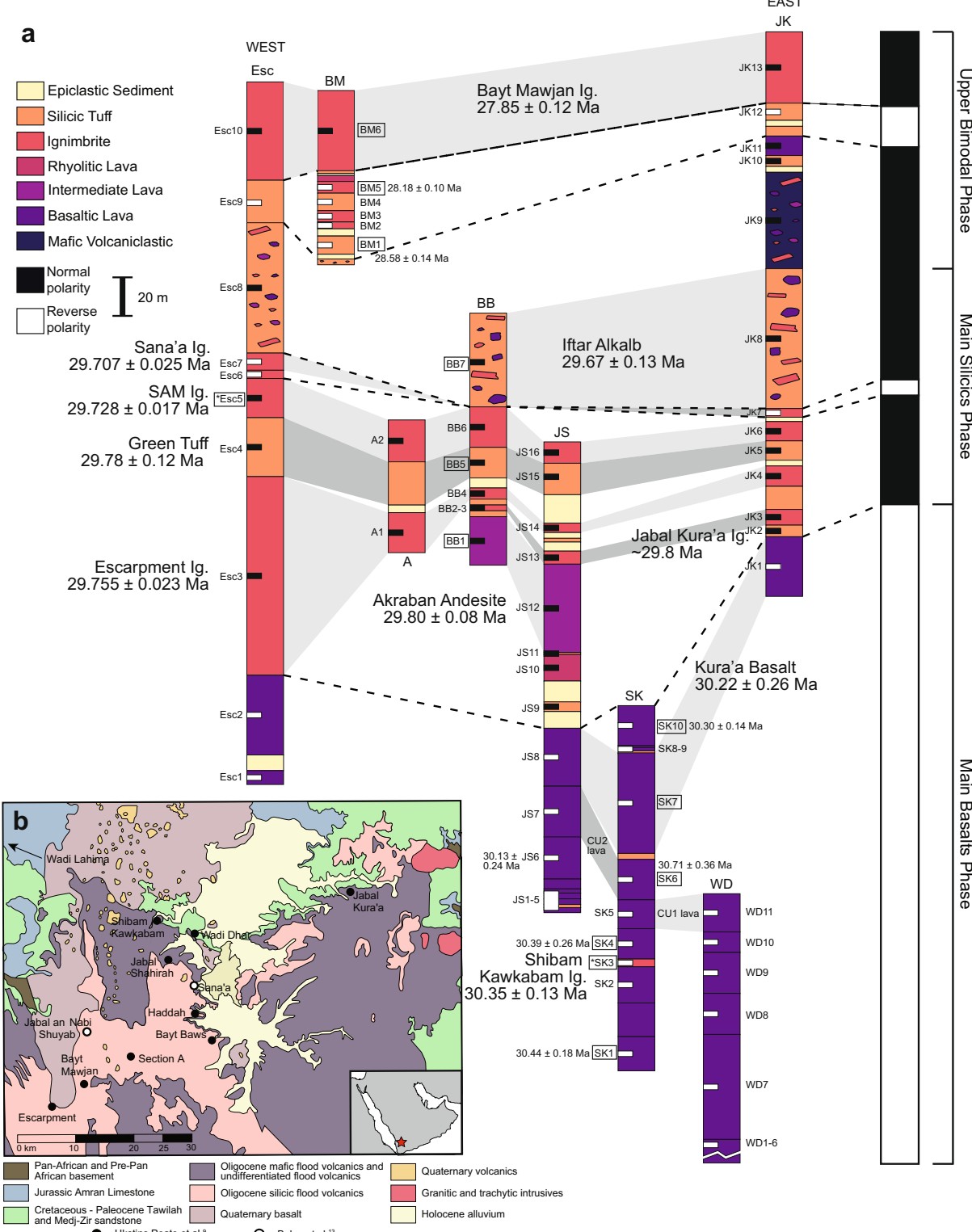

**Fig. 1 Samples profiles of the Northern Yemen volcanic units. a** Schematic volcanic stratigraphy and paleomagnetic sampling profiles of volcanic units emplaced during Oligocene bimodal volcanism in Northern Yemen (modified from Ukstins Peate et al.[9]). Unit thicknesses and lithologies are from Ukstins Peate et al.[9] and paleomagnetic data are from Riisager et al.[8]. Section abbreviations, from west to east, are: Esc Escarpment, BM Bayt Mawjan, A Section A, BB Bayt Baws, JS Jabal Shahirah, SK Shibam Kawkabam, WD Wadi Dhar, JK Jabal Kura'a. Ignimbrite is abbreviated as Ig. Sites are annotated with magnetic polarity data[8] where white and black are reverse and normal polarity, respectively. Sites outlined in boxes denote those dated by $^{40}$Ar/$^{39}$Ar (refs. [8,10,13]) or $^{206}$Pb/$^{238}$U geochronology (data presented here) and ages are shown in detail in Fig. 2. Ages and sites denoted with an asterisk (*) are from correlative units in Ethiopia[10]. Sampling locations are shown in **b**[9] with the Sana'a region, Yemen indicated with a star.

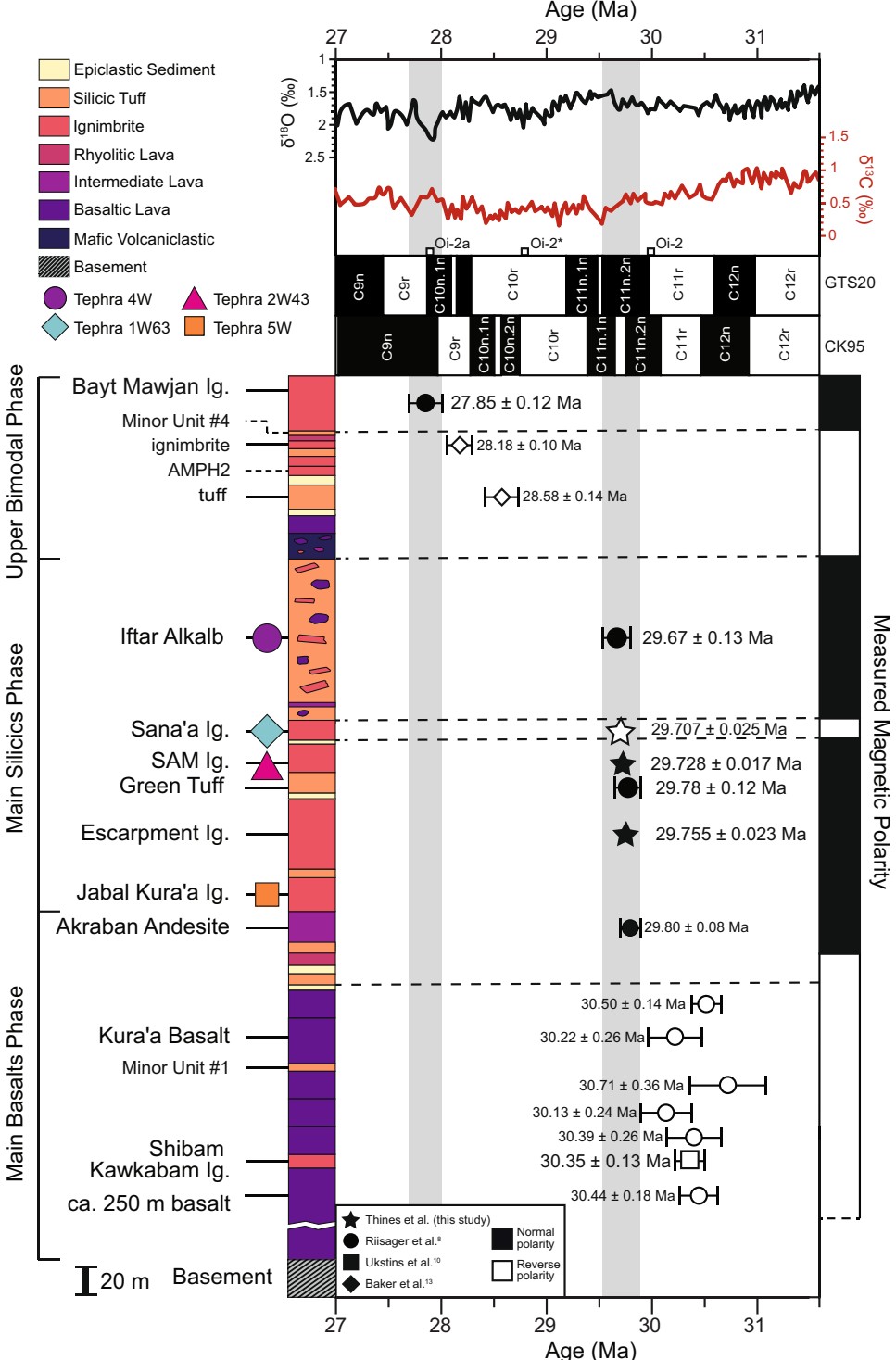

**Fig. 2 Composite stratigraphy of Northern Yemen bimodal flood volcanic units showing paleomagnetic and geochronologic data.** Composite stratigraphy is shown using the average thickness of each unit[9]. Four units have been correlated to Indian Ocean tephra layers[12] and are annotated by the colored symbols. Paleomagnetic data[8] are indicated where white = reverse polarity and black = normal polarity. Dashed lines show the approximate locations of the paleomagnetic reversals in the stratigraphy. Minor Unit #4 and AMPH 2 are from different sample localities and both underlie the Bayt Mawjan Ignimbrite, but their stratigraphic order relative to each other is uncertain. Oligocene flood volcanic units overly sedimentary basement (Tawilah Group sandstone)[9]. Symbols for $^{40}$Ar/$^{39}$Ar ages[8,10,11] are colored based on polarity. The gray field highlights the $^{40}$Ar/$^{39}$Ar and $^{206}$Pb/$^{238}$U ages with associated uncertainties of two pulses of Afro-Arabian silicic volcanism. Error bars for $^{206}$Pb/$^{238}$U and $^{40}$Ar/$^{39}$Ar ages are $2\sigma$ (Supplementary Data and Supplementary Information, respectively). The Escarpment Ignimbrite, Green Tuff, SAM and Sana'a Ignimbrites, and Iftar Alkalb are a set of normal to reversed polarity that encompass the duration of the C11n.1r Subchron and are compared to the GPTS of Cande and Kent[14] as reported in the 2020 Geologic Time Scale[42]. Benthic foraminiferal $\delta^{18}$O and $\delta^{13}$C curves are from the 2020 Geologic Time Scale[42].

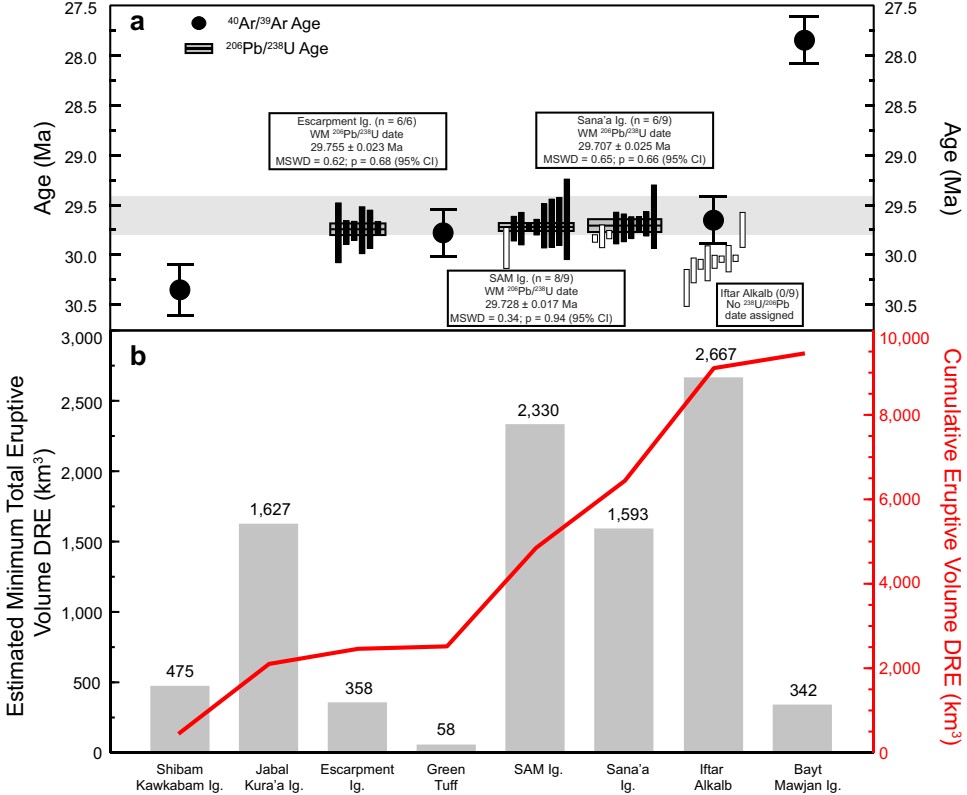

**Fig. 3 $^{40}$Ar/$^{39}$Ar and $^{206}$Pb/$^{238}$U geochronologic data. a** The gray field highlights the ages and associated uncertainties (2$\sigma$) of the Escarpment Ignimbrite, Green Tuff, SAM and Sana'a Ignimbrites, and Iftar Alkalb. Ranked single-zircon and $^{206}$Pb/$^{238}$U dates are shown for the Escarpment, SAM, and Sana'a Ignimbrites. Horizontal gray bars outlined in black indicate the weighted mean $^{206}$Pb/$^{238}$U ages with 95% confidence interval. Error bars for $^{40}$Ar/$^{39}$Ar ages are 2$\sigma$ (Supplementary Information). **b** Minimum total eruptive volume DRE (km$^3$) values are from on-land and correlated deep-sea tephra layers found in Ocean Drilling Program cores from the Indian Ocean, Leg 115[1,9,12].

the observed range of zircon compositions. These findings are consistent with previous modeling of whole-rock compositions of ash shards from correlated deep-sea tephras (Fig. 2), which required a minimum of 60% fractional crystallization of plagioclase, anorthoclase, augite, magnetite, and ilmenite to generate the observed compositional variation[12].

**CA-ID-TIMS geochronology**. Thirty-two grains that showed no sign of inclusions and yielded consistent U-Pb laser ablation dates were plucked from their respective grain mounts for high-precision CA-ID-TIMS geochronology (Supplementary Data). Preference was given to zircon crystals that captured the full range of compositions found in each unit. Six zircon crystals from the Escarpment Ignimbrite yielded a weighted mean $^{206}$Pb/$^{238}$U date of 29.755 ± 0.023 Ma (mean squared weighted deviation (MSWD) = 0.62; Fig. 3). Excluding the oldest zircon crystal from the SAM Ignimbrite (which was older than the $^{206}$Pb/$^{238}$U age of the underlying unit and inferred to be an antecryst), the remaining eight zircon crystals yielded a weighted mean date of 29.728 ± 0.017 Ma (MSWD = 0.34). Six zircon crystals from the Sana'a Ignimbrite yielded a weighted mean date of 29.707 ± 0.025 Ma (MSWD = 0.65; Fig. 3), excluding three zircon crystals older than 29.745 Ma, also inferred to be antecrysts. The weighted mean $^{206}$Pb/$^{238}$U dates have been interpreted as the eruption age of each respective unit. Weighted mean dates for the SAM and Sana'a Ignimbrites calculated with the older zircon crystals are 29.733 ± 0.030 Ma (MSWD = 2.40) and 29.793 ± 0.042 (MSWD = 8.96), respectively.

Although Iftar Alkalb is the stratigraphically youngest unit dated, nine zircon crystals were consistently older (29.731 ± 0.089–30.320 ± 0.094 Ma; Fig. 3) than the weighted mean ages of the other units and so no date was assigned. We attribute this to the emplacement mechanism of the caldera collapse breccia with abundant mega-clasts of underlying stratigraphy contributing xenolithic material or antecrysts that are recording an earlier stage of zircon crystallization. Zircon morphologies (Supplementary Information) and compositions (Fig. 4 and Supplementary Data) were highly variable for Iftar Alkalb and further work is necessary to evaluate these complexities.

**$^{40}$Ar/$^{39}$Ar age recalculations**. Sanidine from the Green Tuff, SAM Ignimbrite, and Iftar Alkalb were previously dated via the $^{40}$Ar/$^{39}$Ar method[8,10]. Those dates have been recalculated using a 28.201 Ma monitor age for the Fish Canyon Tuff sanidine[21]. Recalculations (Supplementary Information) yield a 29.78 ± 0.12 Ma age for the Green Tuff, 29.66 ± 0.14 Ma age for the SAM Ignimbrite, and 29.67 ± 0.08 Ma age for Iftar Alkalb (Fig. 2). Previous $^{40}$Ar/$^{39}$Ar ages[8,10,11] from the Shibam Kawkabam Ignimbrite (30.35 ± 0.13 Ma), Kura'a Basalt (30.22 ± 0.26 Ma), Akraban Andesite (29.80 ± 0.08 Ma), an overlying small-volume rhyolitic tuff (28.58 ± 0.14 Ma) and ignimbrite (28.18 ± 0.10 Ma), and the Bayt Mawjan Ignimbrite (27.85 ± 0.12 Ma) have also been recalculated. The $^{206}$Pb/$^{238}$U zircon ages are in agreement with the recalculated $^{40}$Ar/$^{39}$Ar ages and are compiled and presented here as a revised chronostratigraphy of the Northern Yemen flood volcanics (Fig. 2).

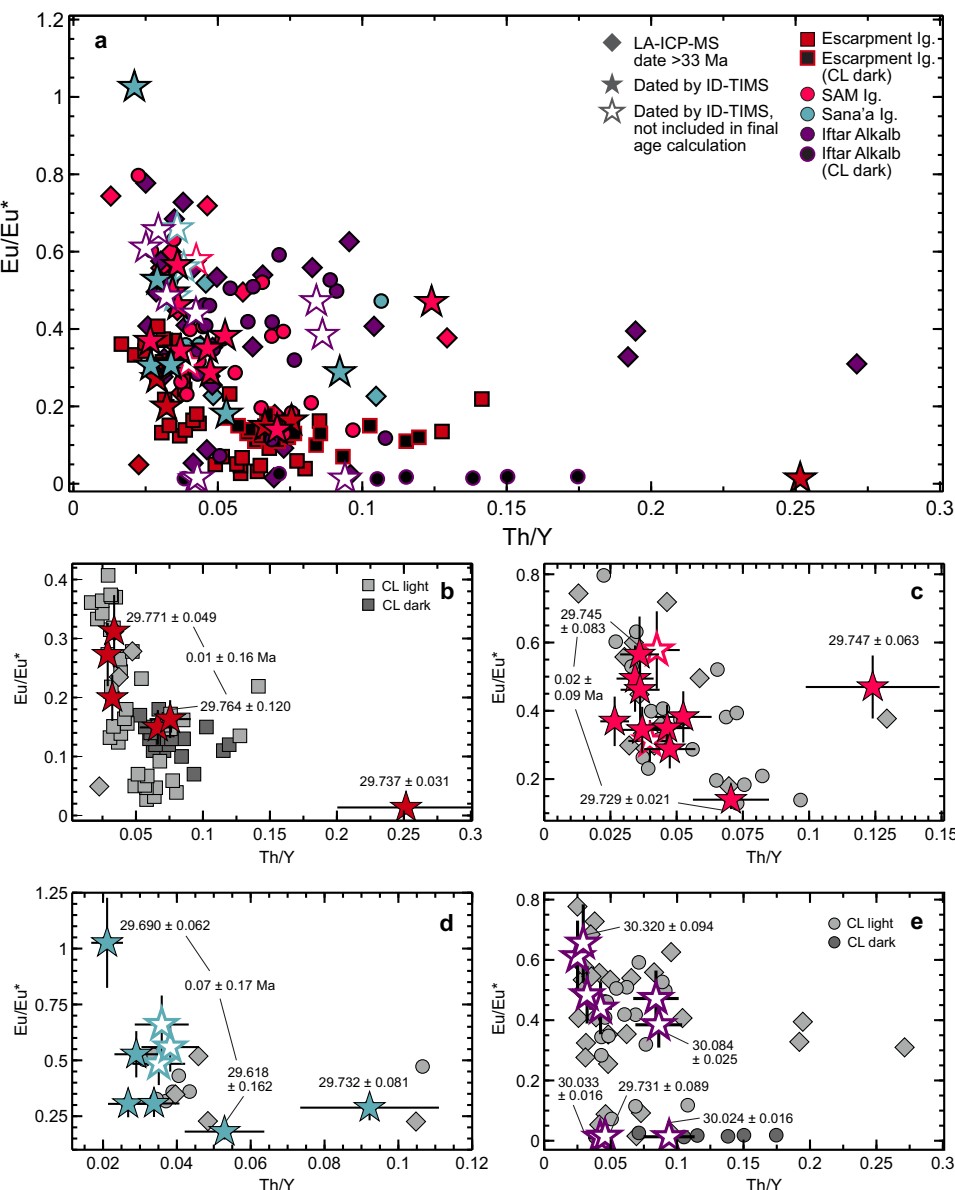

**Fig. 4 Composition of zircon from the Escarpment, SAM, and Sana'a Ignimbrites and Iftar Alkalb. a** Zircon crystals are denoted by age, dating method, and inclusion in final age calculations. Zircons >33 Ma (from preliminary LA-ICP-MS dating, average 2σ uncertainty ±3 Ma) are denoted by diamond symbols. Non-luminescent (CL-dark) zircon crystals from the Escarpment Ignimbrite and Iftar Alkalb are denoted by black symbols. **a–e** show Th/Y versus Eu/Eu* in detail for the Escarpment Ignimbrite (**b**), SAM Ignimbrite (**c**), Sana'a Ignimbrite (**d**), and Iftar Alkalb (**e**). Error bars in **b–e** are 2σ (Supplementary Data).

## Discussion

Elements that are normally incompatible during magma differentiation (e.g., U, Nb, Th, Y, and Hf) and the europium anomaly (Eu/Eu*) in rare earth element patterns resulting from feldspar fractionation are useful indicators of magma differentiation. Assuming that both elements remain incompatible, more differentiated rhyolites will evolve towards higher Th/Y ratios while Eu/Eu* will decrease with continued feldspar crystallization[22]. With a few exceptions, zircons dated via CA-TIMS for these units show the same trend: the least evolved zircon with the highest Eu/Eu* and lowest Th/Y values are older than the most evolved zircon by $0.01 \pm 0.16$ Ma in the Escarpment Ignimbrite, $0.02 \pm 0.09$ Ma in the SAM Ignimbrite, and $0.07 \pm 0.17$ Ma in the Sana'a Ignimbrite (Fig. 4). Thus, ages for zircon crystals spanning the full geochemical ranges are statistically indistinguishable, suggesting that these large volume magmas were rapidly differentiated within $10^3–10^4$ years once the magmas reached Zr

saturation. These estimates are based on LA-ICP-MS single-spot analyses and whole-grain CA-TIMS zircon ages because the small crystal sizes and presence of mineral inclusions made multiple spot analyses difficult. Eu/Eu* and Th/Y are not correlated for zircons in Iftar Alkalb and there is no age relationship between the most and least evolved zircons (Fig. 4), further supporting that the zircons in Iftar Alkalb are of a mixed xenolithic or antecrystic origin.

Magma flux rates (km³/yr) were calculated for 100 and 400 kyr of residence for the Escarpment, SAM, and Sana'a Ignimbrites based on the age difference between the most and least evolved zircon crystals in each unit. For 100 kyr residence, magma flux rates are $3.6 \times 10^{-3}$ km³/yr, $2.4 \times 10^{-2}$ km³/yr, and $1.6 \times 10^{-2}$ km³/yr for the Escarpment, SAM, and Sana'a Ignimbrites, respectively. For 400 kyr residence, magma flux rates are $9.0 \times 10^{-4}$ km³/yr, $6.0 \times 10^{-3}$ km³/yr, and $4.0 \times 10^{-3}$ km³/yr for the Escarpment, SAM, and Sana'a Ignimbrites, respectively.

Upper estimates of $3.6 \times 10^{-3}$–$2.4 \times 10^{-2}$ km$^3$/yr for 100 kyr residence are similar to those calculated for other rapidly assembled large-volume silicic systems (e.g., Yellowstone supereruptions[23,24] and Fish Canyon Tuff[25]). The most conservative estimates using 400 kyr residence ($9.0 \times 10^{-4}$–$6.0 \times 10^{-4}$ km$^3$/yr) are similar to but lower than the minimum calculated magma flux from Yellowstone ($2.8 \times 10^{-3}$ km$^3$/yr for the 280 km$^3$ Mesa Falls Tuff[23]) and significantly lower than that of Taupo (>0.33 km$^3$/yr for the 530 km$^3$ Oruanui eruption[26]).

U-Pb zircon dating shows that three sequential eruptions of Afro-Arabian silicic volcanics—the Escarpment Ignimbrite, the Green Tuff and SAM Ignimbrite, and Sana'a Ignimbrite—were collectively emplaced within a timespan of 48 ± 34 kyr (calculated using the square root of the sum of the uncertainties), yielding a long-term volumetric extrusive rate of $5.27 \times 10^{-2}$–$3.08 \times 10^{-1}$ km$^3$/yr for 4320 km$^3$ DRE. The estimated minimum total eruptive volume for the entirety of the Main Silicics phase is 8620 km$^3$ DRE over a duration of 130 ± 150 kyr, constrained by the ages of the Akraban Andesite and Iftar Alkalb, which yield a lower extrusive rate of $3.05 \times 10^{-2}$–$6.63 \times 10^{-2}$ km$^3$/yr. Northern Yemen unit volume estimates[1,9] are minimum values accounting for the lateral distribution and measured thickness in the studied field areas (Fig. 1) and correlations to Indian Ocean deep-sea tephra layers located >2700 km away from Yemen[12]. While there was wide-scale silicic volcanism following the termination of the main pulse of flood basalt emplacement[13,27], unit volume estimates outside of Northern Yemen remain sparse. Extrusive rates for other regions of the Afro-Arabian province, such as the Ethiopian stratigraphy, are difficult to constrain due to pervasive post-emplacement faulting. Notably, a series of silicic supereruptions in the Tana Basin, Ethiopia[28] have recently been dated at 31.108 ± 0.020–30.844 ± 0.027 Ma with an estimated minimum eruptive volume of 2000–3000 km$^3$, corresponding to a long-term volumetric extrusive rate of 0.8–1.1 $\times 10^{-2}$ km$^3$/yr.

Long-term volumetric extrusive rates of basaltic and andesitic systems are thought to be higher than those of silicic systems by up to two orders of magnitude[4]. Average extrusive rates in silicic systems are calculated to be highest for continental arcs ($4.90 \pm 0.15 \times 10^{-3}$ km$^3$/yr) followed by oceanic arcs ($4.50 \pm 0.79 \times 10^{-3}$ km$^3$/yr), continental rifts ($4.48 \pm 0.86 \times 10^{-3}$ km$^3$/yr), continental hotspots ($1.29 \pm 0.25 \times 10^{-3}$ km$^3$/yr), and continental volcanic fields ($6.47 \pm 1.96 \times 10^{-4}$ km$^3$/yr). The extrusive rate of the Main Silicics phase of the Northern Yemen section of the Afro-Arabian province is most similar to—but notably higher than—the extrusive rates of the central Taupo volcanic zone ($1.28 \times 10^{-2}$ km$^3$/yr; ref. [29]), the silicic portion of Kamchatka ($1.05 \times 10^{-2}$ km$^3$/yr; ref. [30]), and Quaternary phonolites from the Kenya rift valley ($1.20 \times 10^{-2}$ km$^3$/yr; ref. [31]). Our findings are consistent with observations at other large-volume silicic systems that record rapid periods of differentiation and magma reservoir assembly superimposed on lower background fluxes. While some silicic systems have produced more voluminous individual eruptions (e.g., Fish Canyon Tuff with 4500 km$^3$ DRE[32]) and larger cumulative eruptive volumes over longer time intervals (e.g., Paraná-Etendeka LIP with 20,000–35,000 km$^3$ over 6 Myr[33,34]), the eruptions of the Main Silicics phase in Northern Yemen represent the largest long-term volumetric extrusive rate of silicic volcanism on Earth.

Some volcanic provinces appear to coincide with major global environmental change and mass extinctions (e.g., Siberian Traps, Karoo-Ferrar, Emeishan, and Central Atlantic LIPs), yet others, even those with silicic supereruptions (e.g., Paraná-Etendeka LIP), do not[35]. Models for volcanism-driven environmental change predict years of cooling from SO$_2$ injection into the stratosphere from a single eruption and/or tens of thousands of years of warming from CO$_2$ emissions[36]. Several of the Afro-Arabian silicic supereruptions have been correlated to 10–15 cm-thick tephra layers located >2700 km away in the Indian Ocean[12] (Fig. 2), suggesting volcanic fallout on a near-global scale. However, the timing of these supereruptions in relation to several Rupelian-aged cooling events that have been identified in Chrons C12 (Oi1a, Oi1b, and Oi2[37,38]) and C10 (Oi2* and Oi2a[37,38]) indicate that the perturbations in δ$^{18}$O and δ$^{13}$C pre-date the eruptions[16,17] (Fig. 2). Other silicic supereruptions, such as the ~31 Ma caldera-forming eruptions in the Tana Basin[28] and ~28 Ma eruption of the Fish Canyon Tuff[32], likewise do not coincide with global cooling events. The correlation between volcanic eruptions and isotopic perturbations rely on the precision of the eruption ages, resolution of the climate proxy data (±0.2‰)[16], and the sensitivity of the climate proxies to the effects of individual volcanic eruptions. While the Afro-Arabian Main Silicics phase eruptions represent the largest known long-term volumetric extrusive rate of silicic volcanism, they did not cause major global climate change at the current resolution of these data. Challenges remain in discerning the various roles of the tempo, volatile budget, eruption mechanism, and volume of magma extruded from LIPs and their effect on global environmental change. However, robust temporal constraints continue to provide critical insight into this relationship.

Previous efforts have been made to correlate Oligocene Afro-Arabian volcanic deposits with the GPTS[8,39] but those were unable to unambiguously distinguish between the GPTS of Cande and Kent[14] and Huestis and Acton[15]. Recent studies on the Oligocene magnetic polarity sequence have utilized astronomical age models[38], radio-isotope age models[37], recalculations of the Cande and Kent[14] GPTS using updated $^{40}$Ar/$^{39}$Ar flux monitor ages[40], and a combination of all three[37]. One of the lingering issues with distinguishing between an appropriate method for determining the Rupelian age (33.9–28.1 Ma) is the lack of tie points from radio-isotopic dates. The Rupelian/Chattian boundary Global Boundary Stratotype Section and Point records a nearly continuous record of astronomically tuned magnetostratigraphy for the Oligocene but only provides one tie point for the Rupelian for the uppermost Chron C12r with a gap between 31.8 ± 0.2 and 27.0 ± 0.1 Ma[37,41]. The 2012 Geologic Time Scale for the Paleogene[37] favored an integrated radio-isotope, GPTS, and cyclostratigraphy model with sixth-order polynomial fit to produce a complete C-sequence. The C11n.1r Subchron is estimated to have a duration of 0.050 Ma with a −0.654 Ma discrepancy between radio-isotopic and astronomic age models[37]. The only discrepancy between the combined age model of the 2012 Geologic Time Scale and new 2020 Geologic Time Scale for the time range of interest is a shift of the base of Chron C12n to 30.977 from 31.034 Ma[37,42].

We propose that the 29.728 ± 0.017 Ma $^{206}$Pb/$^{238}$U zircon age of the SAM Ignimbrite and 29.67 ± 0.13 Ma $^{40}$Ar/$^{39}$Ar sanidine age of Iftar Alkalb—further constrained to 29.67 ± 0.13 Ma by the 29.707 ± 0.025 Ma $^{206}$Pb/$^{238}$U age of the Sana'a Ignimbrite—can be used as tie points for the GPTS. Our chronostratigraphy and magnetostratigraphy are definitively in agreement with the Cande and Kent[14] GPTS (Fig. 2). Discrepancies between our results and the 2020 Geologic Time Scale arise from the sparsity of radio-isotope dates for the Rupelian coupled with the short duration of the C11n.1r Subchron. Our findings are within the 0.654 Ma discrepancy between the radio-isotopic and astronomic age models and could thus serve as robust tie points for future time scale calibrations.

## Methods
Samples from the Sana'a area of Northern Yemen were previously collected and described in Ukstins Peate et al.[9] (Fig. 1). Paleomagnetic data was measured on 587 oriented drill cores collected at 71 sites[8] (Fig. 1). Zircon U-Pb petrochronology was undertaken at the Boise State University Isotope Laboratory. Zircon crystals from

the Escarpment, SAM and Sana'a Ignimbrites, and Iftar Alkalb were separated using standard magnetic and heavy liquid techniques and annealed at 900 °C for 60 h. Zircons were imaged using a JEOL T-300 scanning electron microscope fitted with a Gatan Mini CL detector and JEOL back-scattered electron detector under 15 kV probe current and 2 mA accelerating voltage operating conditions (Supplementary Information). Trace element analyses and preliminary U-Pb dating for 31–95 crystals per unit (Supplementary Data) were performed using a Thermo-Electron X-Series II quadrupole ICP-MS and New Wave Research UP-213 Nd:YAG UV (213 μm) laser ablation system with a 10 Hz at 5 J/cm$^2$ pulsed laser and 15 μm spot size. NIST SRM-610 and SRM-612 glasses were used as standards for trace element concentrations and Plešovice zircon standard[43] was used for U-Pb calibration. Zircon standards were measured every 10 unknowns; glass standards were analyzed at the beginning of two 109-spot cycles.

A total of 32 crystals from the 4 units were selected for CA-TIMS analysis on the basis of morphology, zoning, chemistry, and preliminary $^{206}Pb/^{238}U$ dates. Zircon crystals were chemically abraded[18] in 120 μL of 29 M hydrofluoric acid (HF) at 180–200 °C for 12 h and then rinsed in 3.5 M $HNO_3$ in an ultrasonic bath for 60 min. The residual crystals were rinsed twice in ultrapure $H_2O$ and transferred to Teflon PFA microcapsules and spiked with ET535 mixed U-Pb isotope tracer solution[44,45]. The spiked residual crystals were dissolved in Parr vessels in 120 μL of 29 M HF at 220 °C for 48 h, dried, and redissolved in 6 M HCl at 180 °C overnight[46]. Pb and U were purified from the chloride matrix using HCl-based anion-exchange chromatography and dried with 2 μL of 0.05 N $H_3PO_4$. High-precision isotope dilution U and Pb isotope ratio measurements were made using a single Re filament silica gel technique on an Isotopx Isoprobe-T multi-collector TIMS equipped with an ion-counting Daly detector (Supplementary Data). Dates are calculated using the decay constants of Jaffey et al.[47]. Analytical uncertainties on dates are reported to be 2σ and propagated using the algorithms of Schmitz and Schoene[48].

## Data availability

Supplementary Information contains cathodoluminescence (CL) images of zircon crystals analyzed by LA-ICP-MS and CA-TIMS and details on the recalculation of $^{40}Ar/^{39}Ar$ ages. Supplementary Data contains details on the LA-ICP-MS trace element concentrations and $^{206}Pb/^{238}U$ dates for zircon crystals dated by CA-TIMS. The full dataset of LA-ICP-MS trace element concentrations for all zircon crystals analyzed in this study are available in the PANGAEA database. Samples collected by I.A.U. are housed at the University of Auckland.

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

## Acknowledgements

This material is based on work supported by the National Science Foundation under Grant Nos. EAR-1759200 and EAR-1759353. Any opinions, findings, and conclusions or recommendations expressed in this material are those of the author(s) and do not necessarily reflect the views of the National Science Foundation. We thank the AGeS program, members of the Boise State University Isotope Geology Laboratory for support with sample preparation and B.D. Cramer for insightful discussions.

## Author contributions

J.E.T., I.A.U. and M.S. designed the research project as part of the AGeS2 Geochronology Program. Sample material was provided by I.A.U. C.W. and J.E.T. prepared samples and analyzed the data with help from M.S. J.E.T. wrote the manuscript with support from I.A.U. and M.S. Figure 1b is modified from Ukstins Peate et al.[9] with additional data from Riisager et al.[8] and used with permission by I.A.U. Progress was overseen by I.A.U, the PhD thesis advisor of J.E.T.

## Competing interests

The authors declare no competing interests.
