## [Peer Review File · Nature Communications]

Volumetric extrusive rates of silicic supereruptions from the Afro-Arabian large igneous provinceReviewers' Comments:

Reviewer #1:

Remarks to the Author:

This is an incredibly well written and organised manuscript with a concise, systematic narrative that provides exciting new insights into the temporal evolution of what is arguably one of the greatest episodes of silicic magmatism in Earth history. I find this manuscript to be impactful for a number of reasons:

- The timing, rates and durations of magmatic processes are fundamental, underpinning knowledge necessary for creating and testing ideas about topics ranging from geodynamics to climate change. Accordingly, these authors have constructed one of the most finely, temporally tuned records for large-scale silicic volcanism anywhere. Such a record and dataset provide key constraints on concepts about volcanism's role in shaping Earth surface environments, including as a calibration standard for understanding processes that require being assessed within robust geochronological frameworks.
- Specific to my latter point above, this manuscript refines the palaeomagnetic time scale during a pivotal time in Earth history, i.e. the Eocene-Oligocene transition that archives Earth's initiation and descent into our current ice age. This will be a great dataset by which to connect and correlate ocean-based records (e.g. ODP cores) with land-based records reliant on palaeomag as a proxy for absolute time calibration.
- A current 'hot topic' is the role of volcanism in altering climate and influencing (generally adversely) societal functioning. The data presented in this manuscript reveals that Earth-system-scale perturbations are not necessarily an inevitable outcome to (super)volcanic eruptions. There is an ever-expanding body of research focussing on documenting and assessing the magnitude of climate change linked to volcanism---to date, evidence seems to indicate that even the most violent recorded eruptions result in hemispherically restricted climatic disturbances that operate on human-scale time scales. Nevertheless, it remains a strongly held view that volcanism is a mechanism that can induce nuclear-winter style climatic response and wholesale global change. This work shows that the timing of the Yemeni eruptions, some of the undoubted largest explosive volcanic events of the Phanerozoic, have little to no correspondence with well documented environmental change deduced from marine stable isotope records spanning early-mid Tertiary time. This is a significant finding for use by the Earth science community interested in understanding drivers of Earth system change on a planetary scale. In fact, the authors have provided an exemplar case study that shows how due diligence and circumspection must be undertaken before too glibly invoking 'volcanic cause' for 'climate-change effect'.
- I am, to use a rather colourful term, 'gobsmacked,' at the rapidity of magmatic differentiation these workers appear to have documented via the evolving REE and incompatible element ratios tied to specific zircons that are resolvable-y different in age. Perhaps this is a well-established truism in petrology circles but to non-petrology-aficionados, who I assume are many such as myself, my understanding of magmatic longevity is that there are proponents who champion magmatic processes that remain operative on 10⁶-7 yr and even longer time scales and another set of proponents who have generated data sets that indicate much shorter durations. This dataset nails down magmatic differentiation occurring over time scales 1000's-10,000's yr and perhaps even shorter. This strikes me as having the potential to inform models and modellers interested in continental crust evolution.
- Lastly, the Yemen region has been far-less studied than its neighbouring African counterparts. The authors have addressed that shortcoming and provide important insights into the geodynamical evolution of a hallmark tectonic region, namely, the Ethiopian-Gulf rift system, a region that researchers worldwide study to gain knowledge about mantle-crust process-response.

For these reasons I consider this manuscript to be apropos for Nature Communications, it would have wide appeal to a broad swath of researchers interested in linking Earth's internal processes to the drivers of environmental change on Earth's surface, it provides fundamental new and exciting data from a key but vastly understudied region for investigating plate tectonic phenomena, and the geochronological and geochemical data the authors have amassed goes far beyond simply accurate and precise dating-rocks-in-Yemen and will be used for studies ranging from volcanism and climate

change, to petrological processes in silicic melts, to calibrating absolute time for a pivotal period of Earth system change. I do have a few minor comments and, once those are dealt with by the authors, I recommend publication.

There is an issue with reference numbering. Starting at about line 156, the numbers appear to be off by 1, i.e. in line 156 ref 18 should be ref 17, in line 156 ref 17 should be ref 16, etc. until the end of the manuscript, including in the Methods section. Also, I think ref 17 in line 155 should be ref 15.

CL should be defined before citing it as an acronym---I think the first mention of CL is line 72

Fig 1. Are the labels for Upper Bimodal Phase and Main Basalts Phase reversed on the figure? In the figure legend (below map) you should give Uskins Peate et al. (2005) and Baker et al. (1996) superscripted reference numbers to tie them to the reference list. Also, define 'Ig' as ignimbrite in the figure caption.

Fig. 2. There are two versions of this figure; the one included with the caption is a bit pixelated and fuzzy but the 2nd version at the end of the manuscript is clear and sharp---make certain that is the one that is used for the published figure. Again, provide superscript reference numbers for the references listed on the figure. Perhaps define what you mean by 'basement' (pre-Oligocene rocks?). Line 443: change "unknown" to "uncertain", this term has a somewhat more 'positive' connotation. Line 449: insert 'in' after 'reported'

Fig. 3. Same comment as above: there are two versions of this figure; the one included with the caption is rather pixelated and fuzzy but the 2nd version at the end of the manuscript is clear and sharp---make certain that is the one that is used for the published figure.

Lines Fig. 4. This is a fascinating result with implications regarding the rate of crystal growth in magmas. If you are correct in identifying these trends as indicative of magmatic differentiation, is it possible to do a back-of-the-envelope style calculation to estimate the amount of feldspar that needs to form in order to reduce Eu/Eu* and increase Th/Y by the amounts shown, and does the mineralogical composition of the successively younger ignimbritic units show a compatible/corroborative increase in feldspar component? One wee issue with the figure: your letter designations on the figure are in lower case but are given as upper case in the caption; these should be consistent.

Review by: Tony Prave, School of Earth and Env. Sciences, University of St Andrews, Scotland

Reviewer #2:

Remarks to the Author:

Editorial Office, Nature Communications

Topic: Review of Thines et al. 2021 (m/s no. NCOMMS-21-16463)

Dear Editorial Office,

This letter accompanies my formal review of the manuscript "'Magma flux of silicic supereruptions from the Afro-Arabian large igneous province" by Dr Thines and colleagues.

The manuscript presents new zircon age and trace element data for a series of supereruptions produced from the Afro-Arabian large igneous province during the Oligocene. The authors find that there is very little time between the main period of volcanism and calculate extreme eruption rates for this period, the highest known on Earth. These findings are exciting, the data are of excellent quality and the manuscript is reasonably well written. CA-TIMS is not an easy method and this is a robust

data set that produces some very nice results. However, I think that some of the terminology is misleading and there are a few points that need to be changed.

I have made detailed comments on the combined manuscript pdf, but also provide some general comments below:

1. Terminology. The main point of the paper is to calculate the ages of these events, which can then be used to calculate how much magma was erupted. The term magma flux is a bit misleading in this case. I've pointed out where I think these types of terms should be changed for consistency.
2. Eruption volume accuracy. These eruption deposits are very large and the estimates of volume are not the precise given the many unknowns like total area, caldera infill and erosion. In many places and in figures these eruption volumes are quoted to the nearest km³. Please address this issue.
3. Introduction. Doesn't really convey why this topic and area is of global interest and importance. The paper calculates the highest ever eruption rate for silicic volcanism (perhaps by an order of magnitude). This section needs to be improved.
4. Discussion. I've highlighted areas that I think need further consideration. Particularly around the calculation of magma evolution. I also struggle to see what the environmental impacts section brings to the paper.
5. Reference order. There are many places where the incorrect reference number is called. This may be an issue with numbers being mixed up. Please check.

Thank you for the opportunity to review, I am happy to be contacted for further comments or reviews.

Kind regards,
Simon Barker

July 21, 2021

Dear Reviewers,

I am grateful for the thoughtful and constructive reviews provided on the manuscript. These comments have significantly helped to improve the quality and clarity of my paper. I have revised the manuscript to address the feedback from the reviewers. Below, I provide point-by-point responses (in italics) to the comments from each reviewer with reference to the original and revised manuscript where appropriate. Basic grammatical and referencing errors from both reviewers have been addressed in the revised manuscript.

Reviewer #1 Comments:

Fig 1. Are the labels for Upper Bimodal Phase and Main Basalts Phase reversed on the figure? In the figure legend (below map) you should give Ukstins Peate et al. (2005) and Baker et al. (1996) superscripted reference numbers to tie them to the reference list. Also, define 'Ig' as ignimbrite in the figure caption.

R: The labels were switched and is corrected in the revised version of the figure. The figure legend has been corrected with the subscripted reference numbers. 'Ig.' has also been defined in the figure caption [line 493 of revised manuscript].

Fig. 2. There are two versions of this figure; the one included with the caption is a bit pixelated and fuzzy but the 2nd version at the end of the manuscript is clear and sharp---make certain that is the one that is used for the published figure. Again, provide superscript reference numbers for the references listed on the figure. Perhaps define what you mean by 'basement' (pre-Oligocene rocks?). Line 443: change "unknown" to "uncertain", this term has a somewhat more 'positive' connotation. Line 449: insert 'in' after 'reported.'

R: There was an error with the way the figures were linked to the manuscript document that caused them to appear blurry. I have re-added the figures to mitigate this but the submitted PDFs are the high-quality versions. The figure legend has been corrected with the subscripted reference numbers. Where present in outcrop, the Northern Yemen volcanic units overly sedimentary basement from the Tawilah Group (Ukstins Peate et al., 2005; doi:10.1007/s00445-005-0428-4). As the basement is not discussed in the manuscript itself, no further detail has been added [line 506 of revised manuscript].

Fig. 3. Same comment as above: there are two versions of this figure; the one included with the caption is rather pixelated and fuzzy but the 2nd version at the end of the manuscript is clear and sharp---make certain that is the one that is used for the published figure.

R: Same as above.

Fig. 4. This is a fascinating result with implications regarding the rate of crystal growth in magmas. If you are correct in identifying these trends as indicative of magmatic differentiation, is it possible to do a back-of-the-envelope style calculation to estimate the amount of feldspar that needs to form in order to reduce Eu/Eu^* and increase Th/Y by the amounts shown, and does the mineralogical composition of the successively younger ignimbritic units show a compatible/corroborative increase in feldspar component? One wee issue with the figure: your

letter designations on the figure are in lower case but are given as upper case in the caption; these should be consistent.

R: Single mode fractional crystallization modelling using the experimental partition coefficients of Rubatto and Hermann (2007; doi: 10.1016/j.chemgeo.2007.01.027) and the fractionating assemblage from Ukstins Peate et al. (2008; doi: 10.1016/j.lithos.2007.08.015) demonstrate that the evolution of Eu/Eu^ in zircon is consistent with previous estimates (Ukstins Peate et al., 2008; doi: 10.1016/j.lithos.2007.08.015) of 50-60% fractional crystallization in these rhyolites [lines 123-129 of revised manuscript].*

Reviewer #2 Comments:

[Line 1] Here and throughout the paper. There is a bit of an issue with mixing terms here. MAGMA FLUX is very different from VOLCANIC FLUX (i.e. eruption rate). This is particularly relevant for silicic systems where large amounts of partial melt stay in the crust. I would suggest changing this title to avoid confusion as most of what you calculate is volcanic flux.

R: The terminology throughout the manuscript has been adjusted such that 'magma flux' is now 'long-term volumetric extrusive rate' and 'short-term accumulation rate' is now 'magma flux rate' to reflect common use in the literature (e.g., Rivera et al., 2016; doi:10.1093/ petrology/egw053; White et al., 2006; doi: 10.1029/2005GC001002).

[Line 12] Here and elsewhere. Be careful with the precision you are claiming with volume estimates. I would say that this is probably +/- 50% so to say eruptions are 2667 km³ exactly is not really appropriate. Maybe round to nearest 100 km³ at least? Might be worth also discussing the uncertainty in these volume estimates?

R: Throughout the manuscript text, estimated minimum total eruptive volumes have been rounded to the nearest 100th for all units except the Green Tuff (~60 km³ DRE) and Escarpment Ignimbrite (~360 km³ DRE). I have also recalculated the long-term volumetric extrusive rates using the rounded volume estimates. Figure 3 still shows the original volume estimates from Ukstins Peate et al. (2005; doi: doi:10.1007/s00445-005-0428-4) and Bryan et al. (2010; doi: 10.1016/j.earscirev.2010.07.001) for reference. A short discussion on the volume estimates in regards to the calculation of the long-term volumetric extrusive rates was added to the 'Long-Term Volumetric Extrusive Rates' section [lines 206-212 of revised manuscript].

[Line 19] By more than 2 orders of magnitude! Maybe express this in km³ per kyr?

R: Volumetric extrusive rates are calculated in km³/yr in agreement with Rivera et al. (2016; doi:10.1093/ petrology/egw053) and White et al. (2006; doi: 10.1029/2005GC001002).

[Line 24] No mention of lack of environmental impacts?

R: Statements regarding the environmental change (or lack thereof) in relation to the Northern Yemen eruptions have been added to the 'Abstract' and 'Background' [lines 22-24 and 81-86 of revised manuscript, respectively].

[Line 25] I think that the beginning of the Background section doesn't really cover the importance of the subject and the relevance of large silicic eruptions (supereruptions) and eruption rate/flux.

R: A statement regarding the importance of long-term volumetric extrusive rates of silicic magmas in the context of LIPs has been added to the 'Background' [lines 34-37 of revised manuscript].

[Line 16] This is a little misleading as none of the 13 Quaternary supereruptions have come from a LIP. This would also be a good place to introduce the definition of a supereruption.

R: While it is true that the many of the recent silicic supereruptions are not from large igneous provinces, the exceptions are the 2.08 Ma Huckleberry Ridge Tuff and 0.63 Ma Lava Creek Tuff from the Columbia River-Snake River Plain-Yellowstone Plateau LIP. The occurrence of LIPs is rare compared to other volcanic events and the Afro-Arabian LIP is among the youngest currently identified (Bryan and Ernst, 2008; doi: 10.1016/j.earscirev.2007.08.008). Even so, the volcanic record is biased towards younger events and some of the main hinderances to the identification of silicic supereruptions are erosion, burial, tectonic fragmentation, and distal ash dispersal (Bryan et al., 2010; doi: 10.1016/j.earscirev.2010.07.001), an attribute that makes this study especially relevant. A succinct definition of supereruptions has been added to the 'Background' [lines 40 of revised manuscript].

[Line 65] Need to clarify why your study is novel and different from these.

R: Perhaps the most notable attribute of the Northern Yemen is the well-constrained stratigraphy that preserves a series of these silicic supereruptions. Previous paleomagnetism and $^{40}\text{Ar}/^{39}\text{Ar}$ studies (Riisager et al., 2005; doi: 10.1016/j.epsl.2005.06.016; Ukstins et al., 2002; doi: 10.1016/S0012-821X(02)00525-3) and correlations to distal tephra layers (Ukstins Peate et al., 2008; doi: 10.1016/j.lithos.2007.08.015) strengthen the robustness of the findings [lines 49-53, and 206-213 of revised manuscript].

[Line 68] I found it a bit of a jump from the last section to here, with no details of the methods employed etc. I know these have been forced to the end in normal nature style but it might be good to have a line in the last sections saying "Here we use high-precision CA-TIMS to refine the timing of eruptions and show that..."

R: I completely agree. Brief statements on the methods have been added to the end of the 'Background' and 'Results' to provide a smoother transition [lines 81-91 of revised manuscript].

[Line 84] First mention on LA-ICPMS. Need to explain why this was done (i.e. to look for crystal inheritance), xenocrysts etc. as mentioned below.

R: I think it is more appropriate to introduce the concepts of xenocrysts and antecrysts during the discussion of the LA-ICP-MS data itself after the context of the zircon morphologies has been given. A brief mention was made in lines 88-91 of the revised manuscript and expanded upon in lines 107-114.

[Line 97] Check number of significant figures in this file for the various trace elements. No point in having REE's to 4 DP.

R: The data in 'Supplemental Information 2' is now reported to the appropriate number of significant figures.

[Line 108] On the basis of CL? How to you decide whether or not to cull these data points? How would these impact on your mean age?

R: The decision was made primarily based on the $^{206}\text{Pb}/^{238}\text{U}$ CA-TIMS zircon ages of the underlying units. The weighted mean age of the Escarpment Ignimbrite is 29.755 ± 0.023 Ma. The oldest dated zircon from the overlying SAM Ignimbrite is older than the 29.755 ± 0.023 Ma age of the Escarpment Ignimbrite and was therefore excluded from the calculation of the final weighted mean date. The same thought process was used for the three oldest zircons from the Sana'a Ignimbrite and all of the dated zircons from Iftar Alkalb. While the inclusion of the single rejected zircon from the SAM Ignimbrite yields a final age that is stratigraphically feasible (29.733 ± 0.030 Ma; MSWD = 2.40), the inclusion of the three rejected zircons from the Sana'a Ignimbrite does not (29.793 ± 0.042 Ma; MSWD = 8.96). Furthermore, the MSWD of the final ages that include the rejected zircons [lines 142-144 of revised manuscript] are anomalously high, further supporting the original approach.

[Line 114] Some of the CL images seem to be distinct? Mention these?

R: In most respects, Iftar Alkalb was an outlier in this study whose complexities are interesting and require additional work to understand [lines 150-152 and 180-183 of revised manuscript].

[Line 118] Perhaps explain why this is important to give context of these recalculations?

R: The main purposes of including the previous $^{40}\text{Ar}/^{39}\text{Ar}$ ages recalculated with the updated monitor age is to demonstrate that the new zircon ages are compatible with the preexisting stratigraphic framework and provide a single place with all of the current dates for the Northern Yemen volcanic units [lines 161-163 of revised manuscript].

[Line 128] Accumulation is the wrong term here. For silicic eruptions accumulation is often used to refer to the build up of a melt dominant body prior to eruption. You are looking at rhyolite magma evolution timescales.

R: 'Short-term accumulation rates' is now discussed in terms of 'timescales of magma differentiation' and 'magma flux rate' to clear up confusion.

[Line 134] I don't agree with this approach. CA-TIMS gives a bulk age on the zircon and is very precise but is an average of the whole grain. You should at least point this out as a limitation here. The LA-ICPMS data are from single points and therefore the age of that particular region of the grain may not match the average TIMS age.

R: Ideally I would have at least one core and rim LA-ICP-MS spot per zircon crystal but the majority of the zircon crystals were either too small for multiple $15\mu\text{m}$ laser ablation spots and/or had too many Fe-Ti oxide and apatite inclusions to obtain clean data [lines 178-180 of revised manuscript].

[Line 137] This is not clear from the figure. Can you plot age vs trace elements if this is your main argument here?

R: The goals of Figure 4 are to show both the evolution of the zircon in terms of Th/Y and Eu/Eu and show the age distribution within each unit. Both could not be achieved by changing one of the axes to age and an additional figure would be redundant. Further clarification has*

been added to Figure 4 by adding the difference in age between the most and least zircon in each unit as discussed in the text [lines 171-174 of revised manuscript].

[Line 140] This is also misleading and has little context. The magmas didn't contain zircon throughout their evolution, it would have only crystallised once the melt reached Zr saturation. You don't have the resolution of U/Th dating as many of the studies you refer to.

R: While there are limitations to the current approach, this is the first study of its kind on the silicic component of the Afro-Arabian LIP and provides evidence for rapid differentiation of the large-volume silicic magmas, an ongoing debate within the geologic community. A discussion of these limitations has been added to tone down the original argument while still presenting the evidence.

[Line 144] What is the basis for using these values? not clear. Is the maximum related to the time since mafic volcanism?

R: These values are based on the age difference between the most and least evolved zircon crystals in each discussed in the previous paragraph - 0.01 ± 0.16 Ma for the Escarpment Ignimbrite, 0.02 ± 0.09 Ma for the SAM Ignimbrite, and 0.07 ± 0.17 Ma for the Sana'a Ignimbrite. These are related to the time since the magmas reached zircon saturation. Previous fractional crystallization modelling of ash shards from the correlated distal tephra (Ukstins Peate et al., 2008; doi: 10.1016/j.lithos.2007.08.015) demonstrate that the rhyolites were generated from extreme fractional crystallization of mafic magma. The evolution of Eu/Eu in these zircons is also consistent with this finding [lines 123-129 of revised manuscript].*

[Line 171] But there is no discussion of why your volume estimates are so good?

R: More information about the Northern Yemen section was added to the 'Background' and reiterated in the 'Long-Term Volumetric Extrusive Rate' section for clarification [lines 49-53 and 206-208 of revised manuscript].

[Line 193] I'm not really sure what this section adds to the paper? There is not really the temporal resolution to assess environmental change as demonstrated in Fig. 2. This section needs to be explained in more detail or dropped as it doesn't really add much beyond previous studies e.g. Prave et al. 2016

R: While I didn't find evidence that these silicic supereruptions resulted in major global environmental change, at least some discussion on the matter is still warranted following the publication of the 2020 Geologic Time Scale and the new $\delta^{18}\text{O}$ and $\delta^{13}\text{C}$ curves (Speijer et al., 2020; doi: 10.1016/B978-0-12-824360-2.00028-0). Eruptions from LIPs (both basaltic and rhyolitic) are sufficiently large to have global impacts yet there is a growing consensus that many did not result in global thermal perturbations (Bryan and Ferrari, 2013; doi: 10.1130/B30820.1). The key to establishing this correlation (or lack thereof) is both high-resolution geochronologic data and climate proxy data. Due to the high number of well-documented cooling events in the Oligocene, this section is more of a cautionary tale of the importance for these high-resolution data and another demonstrating that large volcanic eruptions do not necessarily result in global climate change on scales observable in the geologic record.

[Line 202] Not clear in fig 2 as to the degree of the perturbation. It seems like 29.7 Ma does have a small step but it is hard to tell from the resolution and scale.

R: While there does appear to be small perturbations in both $\delta^{18}O$ and $\delta^{13}C$ at ~29.7 Ma, the current age of the Bayt Mawjan Ignimbrite is too imprecise to make a strong argument for any causality. Figure 2 shows that is a potential area of interest for future consideration and is beyond the scope of this study.

[Line 205] Over long time scales. They would undoubtedly have climate impacts but they are not preserved in the resolution of the climate proxy you show. I don't see how this section adds to the paper and it is not even mentioned in the abstract.

R: The original manuscript was not clear enough about why this section is important and was lacking an adequate discussion on the precision of the climate proxy data. The topic has now been introduced in the 'Abstract' [lines 22-24 of revised manuscript] and 'Background' [lines 75-78 of revised manuscript] with a more thorough discussion in the 'Afro-Arabian Volcanism and Oligocene Environmental Change' section [lines 235-258 of revised manuscript].

Sincerely,

Jennifer E. Thines, Ph.D.

University of Iowa, Dept of Earth & Environmental Sciences

115 Trowbridge Hall

Iowa City, IA 52242 U.S.A.